# Therapeutic Neuromodulation toward a Critical State May Serve as a General Treatment Strategy

**DOI:** 10.3390/biomedicines10092317

**Published:** 2022-09-18

**Authors:** Simon Arvin, Keisuke Yonehara, Andreas Nørgaard Glud

**Affiliations:** 1Center for Experimental Neuroscience—CENSE, Department of Neurosurgery, Aarhus University Hospital, Palle Juul-Jensens Boulevard 165, 8200 Aarhus N, Denmark; 2Danish Research Institute of Translational Neuroscience—DANDRITE, Nordic-EMBL Partnership for Molecular Medicine, Department of Biomedicine, Aarhus University, Ole Worms Allé 8, 8000 Aarhus C, Denmark; 3Multiscale Sensory Structure Laboratory, National Institute of Genetics, Mishima, Shizuoka 411-8540, Japan; 4Department of Genetics, The Graduate University for Advanced Studies (SOKENDAI), Mishima, Shizuoka 411-8540, Japan; 5Department of Neurosurgery, Department of Clinical Medicine, Aarhus University, Palle Juul-Jensens Boulevard 11 Building A, 8200 Aarhus N, Denmark

**Keywords:** criticality, small world, neural network, simulations, neuromodulation, therapy, self-organized criticality, oscillations, brain waves, translational, TMS, tDCS, DBS, VNS

## Abstract

Brain disease has become one of this century’s biggest health challenges, urging the development of novel, more effective treatments. To this end, neuromodulation represents an excellent method to modulate the activity of distinct neuronal regions to alleviate disease. Recently, the medical indications for neuromodulation therapy have expanded through the adoption of the idea that neurological disorders emerge from deficits in systems-level structures, such as brain waves and neural topology. Connections between neuronal regions are thought to fluidly form and dissolve again based on the patterns by which neuronal populations synchronize. Akin to a fire that may spread or die out, the brain’s activity may similarly hyper-synchronize and ignite, such as seizures, or dwindle out and go stale, as in a state of coma. Remarkably, however, the healthy brain remains hedged in between these extremes in a critical state around which neuronal activity maneuvers local and global operational modes. While it has been suggested that perturbations of this criticality could underlie neuropathologies, such as vegetative states, epilepsy, and schizophrenia, a major translational impact is yet to be made. In this hypothesis article, we dissect recent computational findings demonstrating that a neural network’s short- and long-range connections have distinct and tractable roles in sustaining the critical regime. While short-range connections shape the dynamics of neuronal activity, long-range connections determine the scope of the neuronal processes. Thus, to facilitate translational progress, we introduce topological and dynamical system concepts within the framework of criticality and discuss the implications and possibilities for therapeutic neuromodulation guided by topological decompositions.

## 1. Introduction

Neurological disorders have become one of this century’s biggest challenges to social and health care systems, drawing more than 800 billion EUR in annual costs in Europe alone [1]. This is of little surprise, considering the nervous system’s extraordinarily complex arrangement and the neuronal dynamics to which it gives rise and the plethora of ways by which it can go wrong. To address this growing challenge, the development of more effective clinical strategies is urgently necessary. 

Neuromodulation is a method to modulate the activity of distinct regions or cell types of the nervous system. Diverse neuromodulatory techniques have emerged in contemporary times, some of which remain largely within the domains of basic science, such as optogenetic modulation [2,3], while other methods have seen a massive clinical translation. Indeed, among today’s most well-established neuromodulatory treatments are transcranial magnetic stimulation (TMS) for depression [4], deep brain stimulation (DBS) of the basal ganglia to alleviate Parkinson’s disease [5], and spinal cord and peripheral nerve stimulation for chronic pain relief [6].

Neuromodulation therapies have proved especially attractive in pharmacologically resistant disease processes, where drugs are ineffective or have unacceptable adversities. Using targeted approaches, such as stereotaxic DBS, or MR-guided focused ultrasonography (MRgFUS), neurosurgeons are provided ways to delicately modulate, or ablate, distinct neuronal populations while sparing the broader organism from unintended strain. Pain relief through spinal cord neurostimulation, for instance, represents a remarkably effective treatment for chronic pain syndromes, substantially reducing the need for opiates and the risk for addiction [7]. Similarly, essential tremors have recently become treatable by MRgFUS-thalamotomy, sparing patients from the traditional beta-blocking regime that carries significant risks to cardiopulmonary physiology, e.g., bradyarrhythmias and respiratory insufficiencies [8].

Translational neuroscientists are expanding the medical indications for neuromodulation therapy. Increasingly, researchers are adopting a holistic approach to nervous system pathology aligned with theories of systems neuroscience. Striking examples include the rehabilitation of consciousness through peripheral vagus nerve stimulation [9,10], or thalamic stimulation [11,12,13], the alleviation of schizophrenia via prefrontal TMS [14], or transcranial direct current stimulation (tDCS) [15], and the restoration of memory performance through DBS of the medial septal nucleus [16]. Common to these examples is the strikingly non-linear global effects caused by an otherwise localized stimulus, a phenomenon called diaschisis [17], which is characteristic of a multidimensional dynamical system [18].

Systems neuroscientists put forward the idea that neurological disorders, such as epilepsy, at the most profound level spring from deficits in the systems-level structures and rules, such as those governed by the neural network’s functional topology and brain wave coherence [19,20,21]. Indeed, there is a growing consensus behind the hypothesis that brain waves are instrumental to the brain’s functional architecture [22,23]. In support of such a fundamental role, brain waves have been associated with an extraordinary diversity of neurological processes ranging from memory formation and spatial and cognitive navigation in the hippocampal formation [24,25], to sensory perception and consciousness throughout the cerebral cortices [26]. It has been suggested that neuronal communication between distributed neuronal regions rapidly and flexibly forms and dissolves again based on the distinct patterns by which neuronal population activities coincide [27].

Akin to a fire that may spread or die out, the brain’s ever-changing activity may similarly hyper-synchronize and self-amplify, such as epileptiform seizures, or dwindle out in dissonance and go stale, as in a state of coma. Yet remarkably, the healthy brain remains narrowly hedged in between these dynamical extremes in a so-called “critical state” around which neuronal activity maneuvers local and global modes of operation [28,29]. It has been suggested that perturbations of this brain system criticality underlie distinct neuropathological presentations, such as disorders of consciousness (DoC), epilepsy, and schizophrenia [30]. However, despite the obvious clinical potential, the theory of criticality has yet to make a major translational impact in neurological care. 

Recent computational findings produced in our laboratory demonstrate that a neural network’s short- and long-range connections have distinct roles in sustaining the critical state [31]. While short-range connections shape the dynamics of neuronal activity, long-range connections determine how far this activity spreads and thus ultimately determine the scope, or state, of neuronal processes. This insight makes critical systems theory tractable as a general therapeutic strategy for neuromodulation by meriting the targeting of distinct components of neural network topology. To facilitate translational research in this vein, it is necessary to bridge the gap between theory and clinical practice. To this end, we introduce topological and dynamical systems concepts within the framework of criticality before discussing the implications and possibilities for neuromodulation therapies. Due to the inherently systemic scope of critical dynamics, this article’s focus is on disorders that affect the systems-level neuronal activity, such as DoC and epilepsy.

## 2. Neural Network Topology and the Critical State

How neuronal nodes interconnect have profound consequences on how information is processed by the neural network [32,33]. Indeed, a network can be wired in a multitude of ways, all of which ultimately define the nature of its topology (Figure 1A). Consider the case where each node connects plainly to its nearest *k* neighbors. Here, all areas of the network become identical by forming a lattice of the same repeating pattern: The network is said to be perfectly ordered. In turn, by wiring all the network’s connections at random, we define the opposite extreme, which is said to be perfectly disordered.

Graph scientists quantify such arrangements by the extent to which the nodes cluster (transitivity), and the extent to which the nodes are separated (average shortest path length) (Figure 1B). Accordingly, the fully ordered network contains segregated clusters of nodes that hold few to no long-range interconnections. Contrastingly, the disordered network is marked by promiscuous nodal dispersions that sparsely, if at all, form real clusters. Between these extremes, we find a critical arrangement that retains the clustering of ordered networks, while providing a low nodal separation by the virtue of long-range short-cuts interconnecting distant parts of the network. In their seminal 1998 paper, Watts and Strogatz dissected this semi-random arrangement which they called “small-world” in reference to Stanley Milgram’s experiments on social networks and the six degrees of separation phenomenon [34]. The small-world system is said to be at a point of criticality, referring to the state of a system near a qualitative transformation—a marked shift in how the system behaves. Here, it attains traits that are unique to the states that bound it (Figure 1).
Figure 1The small-world topology. (**A**) By randomly rewiring an ordered lattice, it transitions into a disordered graph. Through this transition, the small-world arrangement defines a critical state. *x* and *y* demarcate two arbitrary nodes on the graph, connected by a red line through the shortest path length between the nodes. (**B**) With increasing disorder, or randomness, the separation between the network’s nodes rapidly decreases (red), while clustering remains practically unchanged (green). The small-world topology corresponds to the mid area marked by high clustering and low separation properties. The *x*-axis is logarithmic. (**C**) Criticality defines the state of a system undergoing a phase transition, a classic example being a sandpile on the verge of collapse [35]. As sand is poured onto the pile, it tends to the critical state hedged between stability and ruin before finally collapsing. Despite the quantity poured onto the pile, it keeps orbiting the critical point, cycling through the same three phases: stability, criticality, and collapse. This is a special type of system behavior, which the brain is thought to possess, called self-organized criticality (SOC). Here, the system, by intrinsic means, tends to the critical state. Figure adapted from the authors’ previous work [31], and the original depiction of small-world graphs by Watts and Strogatz [34].
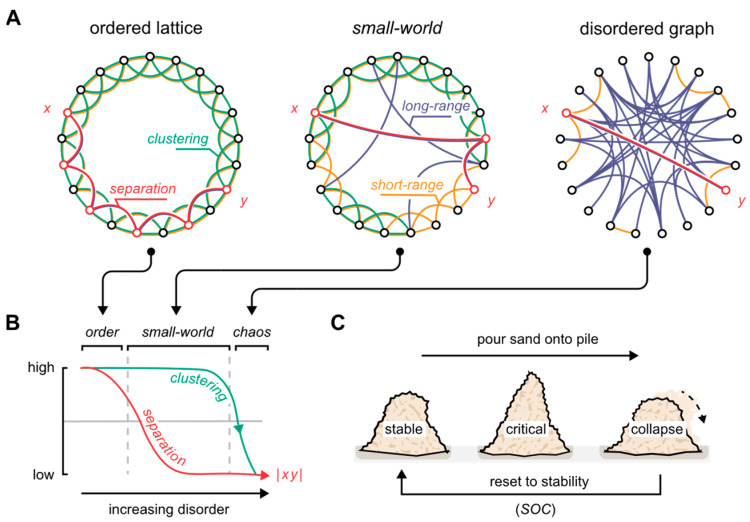


What effects do the structure of a network have on the signals that propagate within it? Work has shown that, as the topological separation increases, signals tend to segregate into local clusters due to the resistance to global transmission exerted by long path lengths [28,32,33]. When nodes are more promiscuously interconnected, however, the signals begin to spread beyond the local span, ultimately saturating the global network activity. Intermediately, near the small-world criticality, the signals are more likely to reverberate through local and global patterns of activity [36,37], akin to the fluctuations of brain activity through different scales of space, in one moment largely regionalized, while, in the next, spanning global interacting assemblies [22]. In other words, maneuvering the critical state seems to provide the means for the brain to tune its operational mode to local or global computational scopes on demand. As Cocchi and colleagues succinctly note [29], “brain function does not only rely upon the execution of particular functions but also on adaptive switching from one function to another based on context and goals.” It has thus been widely hypothesized that the brain’s topology bears resemblance to small-world architectures [38].

Systems poised near criticality possess several computational advantages, such as a high dynamic range, an efficient information capacity, and a high information transfer fidelity [28]. Aberrations from the critical state could, therefore, spur systems-level brain disorders, e.g., insomnia, schizophrenia, and epilepsy [30,39,40,41]. Within this framework, seizures represent states of global neuronal hyper-synchrony, or super-criticality [42], while schizophrenia aligns with a type of sub-critical network dynamic in which long-range neuronal communication is poorly utilized, causing neuronal activity to remain weakly coordinated [43].

Similarly, a DoC constitutes a pathologically sub-critical network state marked clinically by a prolonged deficit in consciousness. More formally, DoCs are divided into the minimally conscious state, the vegetative state, and the comatose state, in ascending order of severity, corresponding to an increasingly weak and segregated nature of neuronal activity [44]. Examining the human brain as a system operating near a state of criticality could thus provide important clues on the nature of brain pathology and potential future treatments [30].

In theory, systems operating at criticality will stay critical indefinitely, if left undisturbed. In practice, however, the critical state resembles an unstable equilibrium, in the parlance of dynamical systems theory, in which noisy disturbances and stochasticity inherent to any naturalistic system inevitably push the system toward extreme states. Based on this inclination, Hesse and Gross argue that the brain must possess mechanisms to sustain and maneuver the critical state—A neuronal implementation of self-organized criticality (Figure 1C) [36]. We will return to this concept in Section 4.

Neuronal activity inherently constrains to the anatomical medium through which it propagates, and by which it is generated. Reciprocally, neurons act onto their medium, too, fluidly shaping it over long and short timescales, such as through neuroplasticity [45,46,47], and brain wave coherence [20,27]. How neuronal populations interact in space and time affects how the activity flows through the network [27]. The anatomical wiring is essentially filtered by the collective neuronal process to effectuate some connections, while abandoning others, forming a functional connectome that tends to a critical arrangement [38,48,49]. In the following section, we summarize how brain wave interactions provide one fundamental method for the continuous reshaping of the neural network’s functional topology.

## 3. Brain Waves and Functional Topology: A Neuronal Communications Syntax

Bishop was among the first experimentalists to report that neurons oscillate through high and low states of excitability [50,51]. At the cellular level, this manifests as subthreshold oscillations of the membrane potential that pose the neuron in a graded state of excitability (Figure 2A, inset). These cellular rhythms make the timing of when signals are sent and received crucial for how well neurons communicate: When neuronal activity resonates, it facilitates the selective transfer of information from one neuron to another (Figure 2A) [52].

The resonance tendencies of population neuroelectric oscillations may, in a similar way, direct the flow of neuronal information (Figure 2B). “Brain waves” refers to the oscillatory neuronal activities that can be observed in virtually all neuronal populations in the mammalian brain [53]. The mechanistic source of these oscillations is thought to be elementary circuit motifs, including local excitatory, inhibitory/principal cell, interneuron interactions, or gap junctions [54]. Brain waves are contained within the electric field potential that represents the spatially integrated electric current flow of hundreds to millions of neurons, depending on the location and size of the measuring electrodes: High spatial resolution is provided by intracranial (electrocorticographic, ECoG) or intracerebral probes, whereas a lower spatial resolution, with a wide integrative window, is offered by scalp electrodes (electroencephalogram, EEG).

Fundamentally, brain waves are similar to single-neuron subthreshold oscillations in the sense that they reflect the cycling of neurons through varying levels of receptiveness. This centers on the premise that neuronal communication relies on some degree of oscillatory coherence, that is, a complete or partial synchronization of distinct neuronal parameters, such as excitation, inhibition, or receptiveness (“input gain”). Accordingly, “communication through coherence”, a model pioneered by Fries [55,56], presupposes that neuronal inputs consistently coincide with moments of high input gain in the post-synaptic receiver. When signals arrive at peak receptiveness, the underlying circuit motifs are activated, which, on the one hand, pass the input onward to downstream receivers, while, on the other hand, closing the gate on subsequent inputs, such as through strong local inhibition. This makes neuronal communication pulsatile because information is transferred only during narrow periods of the oscillatory cycle. Thus, the timing of the signal transmission becomes instrumental in establishing neuronal communication and, consequently, in shaping the functional network topology.

Brain waves occupy an extensive spectrum of frequency bands that have been correlated to distinct neurological processes, such as perceptive grouping through cortical gamma (30–100 Hz) [57], spatial and cognitive navigation through hippocampal theta (6–10 Hz) [24], and memory consolidation through sharp-wave ripples (alternating 5–15 Hz sharp-waves and 150–200 Hz ripples) [58]. Notably, the frequency distribution of brain waves obeys a non-integer power law, which prevents rhythms from perfectly interlocking: This facilitates a metastable system dynamic fostering perpetual fluctuations between stable and unstable brain states, consistent with a system operating near criticality [59].

Thus, both the dynamical and topological features of neural networks are poised to retain proximity to a critical state. Indeed, through the rapid oscillatory interactions of the distributed neuronal assemblies, the brain’s functional connectivity rapidly reorganizes in accommodation to ongoing neurological objectives. It has been argued that these reorganizational processes approach a critical topological state coinciding with a small world-like architecture [60,61], and that perturbations away from this criticality should manifest as systems-level neuronal dysfunction [28,30]. In the next section, we address how this might unfold, and how it can shape new neuromodulation strategies.

## 4. Reestablishing Critical System Dynamics: A General Strategy for Neuromodulation

In the first two sections of this hypothesis article, we reviewed the literature suggesting that brain waves interact to shape the brain’s functional topology around a critical state according to changing neurocomputational demands. Theoretical work, supported by empirical reports, indicates that this reorganizational process is anchored near a topological criticality that coincides with the small world architecture [60,61,62,63]. We now examine specifically how neural network topology and critical dynamics might combine to spur human neuropathology. From here, we dissect the rationale for therapeutic neuromodulation strategies guided by topological decompositions.

As previously described, simulations show that neural networks initiated outside a critical state, or without the means to maintain it, deterministically terminate in extreme equilibria due to the inherent quenching or amplification of neuronal activity that is driven by the network’s topological predispositions [36]. Accordingly, sub-critically inclined networks terminate near null activity states, akin in human patients to comatose- or sleep-like brain states with sparse to no global activity patterns [64,65,66]. Super-critically inclined networks, in turn, cycle through states of hyper-synchrony and intermittent neuronal refraction, corresponding in effect to ictal seizures and interictal pauses [42,67].

Such a tendency to deviate from criticality applies especially to the biological neural systems. Indeed, due to their inherent noise and stochasticity, biological neural networks will inevitably terminate near non-critical extremes if provided with no mechanisms to retain a state of criticality [36]. In a string of pioneering work, Bak dissected the concept of self-organized criticality (SOC), an emergent system phenomenon that relies on the continuous tuning of intrinsic control parameters to approach a critical state (Figure 1C) [36,68]. In line with the idea of SOC, we argue that brain disorders, e.g., caused by neuro-trauma or erroneous neurodevelopment, could be approached analytically as a matter of imbalances between the capacity of the SOC mechanisms to retain brain criticality, and the magnitude of system propensities attempting to break it (Figure 3 and Figure 4). 

In our recent computational article [31], we decomposed the oscillatory activity of artificial neural networks based on the spatial length between the interacting neuronal nodes. This decomposition revealed, first, that short-range connections modulate the dynamics of the system by, in effect, buffering its inherent topological tendencies. Second, long-range connections were found to be the principal defining component of the system state itself, thus satisfying the role of the neural network’s SOC control parameter (Figure 3A,B). Phrased differently, short-range connections modulate the system’s stability, whereas long-range connections determine how far neuronal activity ultimately spreads. The implication is that brain system imbalances, in terms of critical operation, may emerge from disturbed short- and long-range neuronal interactions, respectively. Below, we consider the neural network effects of each topological domain separately, and finally in combination.

### 4.1. Short-Range Neuronal Connectivity Affects the System’s Stability

Simulations show that the volatility of a neural network, especially near criticality, increases as the network’s short-range connectivity deteriorates (Figure 3C) [31]. The same seemingly applies to human brain disorders that have a significant cortico-dysfunctional component, such as schizophrenia [43,69], autism [70], cortical dysplasia [71], and spastic cerebral palsy [72]. These disorders are distinguished by sub-critical characteristics in the form of weak, segregated brain activity [73]—a deficit in cognition and in the engagement of global network assemblies [74]. Yet, paradoxically, cortico-dysfunctional disorders are closely associated with epileptic seizures as well, i.e., a pathologically super-critical state of global neuronal hyper-synchrony [75]. Such a concurrency of sub- and super-critical manifestations fundamentally aligns with a destabilization of the critical state that favors extreme topological equilibria. 

Theory predicts that treatments that augment short-range neuronal connections should stabilize the network’s system dynamics too (Figure 4C) [31]. In line with this prediction, TMS of cortical gamma oscillations—which are thought to be biophysically inclined to local computations [19,76]—has been shown to alleviate schizophrenia in human patients [14]. Similarly, tDCS significantly improves the working memory in schizophrenic patients, coinciding with an increase in cortical gamma synchrony [77]. Analogous effects have been produced in mice by modulating the activity of cortical GABAergic interneurons [78], which are instrumental to the pacing of the gamma cycle [57]. However, considering the significant overlap in epileptic and schizophrenic patients, it is striking to note that no study has yet, to our knowledge, tested the correlation between the resolution of schizophrenic symptoms and a reduction in epileptic events. It is certainly plausible that the dysfunction of the gamma circuitry—hence, short-range neuronal connectivity—is instrumental to both epileptogenesis [79], and schizophrenia [80]. 

A corresponding pattern is provided by vagus nerve stimulation (VNS), a treatment where electric pulses are applied to the peripheral vagus nerve to alleviate drug-resistant epilepsy [81]. Specifically, VNS is known to evoke widespread cortical synchronization [10,82,83], which, if one accepts the premise that excessive long-range neuronal synchrony spurs seizures [31,67,75,84], should be epileptogenic, and not anti-convulsant. Adding to this discrepancy, several studies have shown that VNS elevates arousal and cortical excitability acutely [10,85], and even chronically to the point of rehabilitating the minimally conscious patient [9,86]. Thus, similar to cortico-dysfunctional disorders, VNS affects neuronal activity in a seemingly paradoxical manner, which may be explained by the effects on the underlying system dynamic. Indeed, a potential vagal pathway for the modulation of the system dynamic (via short-range neuronal connectivity) is the control of cortical gain through coerulo-cortical noradrenergic projections [87,88]. Congruently, research shows that both acute and chronic lesions to the locus coeruleus greatly diminish the anti-convulsant effects of VNS [89], and facilitate status epilepticus [90], signs that together are suggestive of a system destabilization. From a translational point of view, these findings indicate that the vagus nerve could be utilized as a peripheral point of access for modulating brain system dynamics. This aligns with an emerging corpus of evidence that supports VNS in the treatment of refractive depression and in the post-traumatic rehabilitation of consciousness [9,86,91,92], two archetypical sub-critical system presentations.

### 4.2. Long-Range Neuronal Connectivity Defines the System’s State

Long-range neuronal connections determine the system’s state, meaning that a state of criticality could be maintained and maneuvered by the modulation of the network’s long-range connectivity [31]. Disturbed long-range connections could therefore produce neuropathologies marked by critical state deviations; Indeed, low levels of long-range connectivity quench global neural network interactions (sub-criticality), whereas excessive levels spur hyper-synchronous seizures (super-criticality). Consequently, a significant disruption of the brain’s long-range connectivity should be expected to shift the brain toward a sub-critical state, i.e., an acute arrest of global network activity (Figure 4B). Intuitively, this should manifest as the loss of consciousness [65,93]. Aligned with this intuition, traumatological reports show that the extent of diffuse axonal injury to the brainstem, but not the cortex, correlates with the immediate loss of consciousness, and the persistence of post-traumatic coma [94,95,96]. This finding has been attributed to lesions within the brainstem’s reticular activating system that modulates thalamic nuclei responsible for long-range cortico-cortical synchronizations via thalamo-corticothalamic loops [97,98,99]. Damage to the thalamus, or its neuronal prerequisites in the brainstem, could therefore disrupt long-range neuronal connectivity through impairments to global neuronal coherence (see Section 3) [100]. This agrees with the fact that vegetative and comatose patients most consistently have damage to the thalamus, or its relays, but not to the cerebral cortex itself [101,102]. Similarly, it has been shown that severely disabled, yet conscious, patients tend to have aggregate focal injuries that, however, consistently spare the thalamus [103]. Yet, despite the consistent association of thalamic lesions with DoC in humans [100], the thalamus may not be necessary for consciousness after all since its full ablation fails to cause unconsciousness, at least in a rodent animal model [104,105]. This merits further examination of the properties of specific brain regions, such as the thalamus and basal ganglia, in neuronal diaschisis and global network connectivity.

It is interesting to entertain the idea that a minimum level of long-range connectivity, hence global neuronal interactions, is necessary to sustain consciousness [106]. As this threshold is breached, the mechanisms of SOC may in effect be void, leaving the brain state unable to escape the sub-critical/unconscious state to reach criticality (Figure 4B). Consistent with this line of thought, thalamic neuromodulation’s efficacy to restore consciousness fundamentally depends on the severity and duration of unconsciousness, perhaps secondarily to a loss of residual regenerative capacity [107]. Accordingly, DBS of the thalamus is generally reported to produce limited behavioral improvements in patients with severe DoC, worsening as the delay before treatment increases [108]. A similar pattern is observed for amantadine [109], a neurostimulant that aids global cortico-thalamic loop interactions by releasing thalamic nuclei from pallidal inhibition [110,111]. By contrast, patient outcomes in mild to moderate DoC have been strikingly positive [112,113,114,115], congruent with the premise that fruitful rehabilitation hinges on a sufficient reserve of long-range neuronal connections to reach and maneuver a state of criticality.

### 4.3. Merits of Combinatorial Neuromodulation Strategies

The decomposition of neural network activity into static and dynamic drivers invites the testing of combinatorial neuromodulation strategies targeting both the short- and long-range topological domains of the network. Indeed, the synergic potential of differential topological modulation, as suggested by computational analyses, is supported by the distinctive pattern of neuronal hyper-connectivity that is often observed in the brain after neuro-trauma. Such hyper-connectivity has been hypothesized to reflect a compensational mechanism to injury [116,117,118,119], which, we suggest, may act to restabilize the critical state through enhanced short-range neuronal connections [31]. Loyal to this idea, the potentiation of cortical excitability through TMS [120], and long-term VNS [10,86], has been shown to significantly improve coma recovery scores in moderate DoC.

Combinatorial neuromodulation research has mostly focused on anti-depressant therapy [4,121,122], an indication for which it has proved remarkably effective. [92,123,124]. However, only sparse literature exists outside anti-depressant objectives; Among the few, a recent pilot study by Bender Pape and colleagues revealed that the behavioral gains in patients treated for DoC doubled when TMS preceded the provision of amantadine [125]. It is necessary for future research to extend upon this work and more broadly test combinatorial strategies for neuromodulation outside conventional therapeutic arenas. The strategy proposed here is, essentially, to stabilize the system dynamic through short-range neuronal potentiation, e.g., by TMS [10,14], tDCS [15], or VNS [86], allowing for adjuvant long-range potentiation, e.g., via DBS of thalamic nuclei [97,98,107], to sustain criticality more easily. Such a combinatorial paradigm could be especially helpful in the treatment of post-traumatic DoC where combined short- and long-range neuronal injuries more likely are the cause pathology.

We have thus far focused on the rehabilitation of pathological brain states and brain dynamics. It is important to note also that topologically guided neuromodulation could have a utility outside conventional therapeutic objectives. A deliberate destabilization of the brain’s system dynamics, for instance, could be used to potentiate the effects of general anesthesia by facilitating sedated/sub-critical system states. This could reduce the need for high-dose anesthesia, and, consequently, the risk for complications, such as cardiopulmonary failure [126,127], and postoperative urinary retention [128]. Moreover, integrating the framework of topological decompositions into deep neural network architectures could enable the development of advanced, closed-loop neuromodulation technology that adapts to ongoing neuronal activity. To these ends, future work must probe the neuroelectric effects of topologically differentiated neuromodulation in more granular detail, such as through high-density ECoG.

In a similar vein, approaching neuro-trauma from a critical systems perspective could advance acute management strategies. As noted previously, neural networks with impaired short-range connectivity are unstable and thus favor extreme system states, especially within subcriticality [31]. In severe cases, this could trap the brain in a persistent state of unconsciousness due to the SOC mechanisms essentially being overcome by excessive topological propensities (Figure 4C). We speculate that such a trapped state could worsen or become functionally irreversible through subacute neuroplastic changes that depress general synaptic gain, such as via Hebbian rules [47]. This challenges the merits of neuroprotectants that prolong central nervous system depression, e.g., hypothermia [129], and induced coma [130], instead lending support to expedited rehabilitation [131]. and neurostimulant therapy [132]. Congruently, a 2011 Lancet study reported that hypothermia may be ineffective, or harmful, in severe brain injury [129], and while induced coma has been shown to improve immediate mortality [133], several studies have shown that prolonged anesthesia disrupts synaptic architecture and causes chronic cognitive deficits [134,135]. Randomized controlled trials, however, are lacking, which is not surprising considering the ethical dilemmas of acute neuro-traumatic care. Nevertheless, it is necessary for future research to elucidate the effects of neuro-trauma on critical system dynamics to test methods improving not only acute mortality but also long-term patient outcomes. Quantitative EEGs could here provide a useful tool for the continuous assessment of patient neuroelectric status [136].

## 5. Conclusions

In conclusion, this hypothesis article has highlighted the relationship between neural network topology and the neuronal dynamics that pervade it. Specifically, we argued that, through short- and long-range neuronal synchronizations, the brain’s functional topology maneuvers a critical state to optimize and adapt its neurocomputational scope continuously. Ample empirical evidence supports the premise of a functional division of network topology based on the length of neuronal connectivity, but causative data are needed, namely, to dissect neuronal responses to neuromodulation from a critical systems perspective. Given the potential role of differential neuromodulation in the treatment—and prophylaxis—of systems-level human neuropathology, future research should be dedicated to advancing our understanding of topological system decompositions.

## Figures and Tables

**Figure 2 biomedicines-10-02317-f002:**
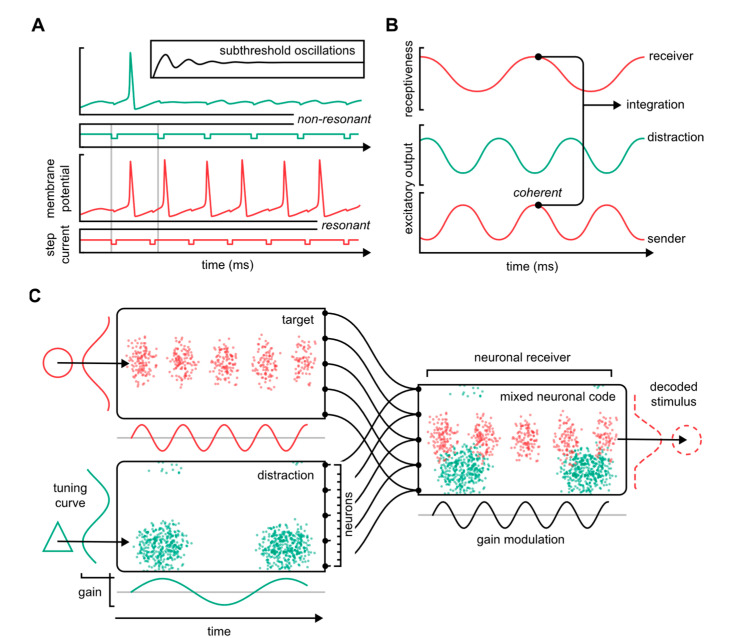
Neuronal dynamics. (**A**) Resonance of the neuronal membrane potential facilitates the selective transfer of information. The traces show a Hodgkin–Huxley neuron receiving step current pulses at two different frequencies (green: low frequency—red: high frequency). When the pulses do not resonate with the neuron’s underlying resonance tendency, the signals are poorly integrated (green). When pulses do resonate, the neuron integrates and transmits the signals onward (red). The inset shows spontaneous subthreshold oscillations. (**B**) In the “communication through coherence”-framework, neuronal inputs systematically arrive at moments of high receptiveness, or input gain. (**C**) Oscillation-based neuronal communication differentiated by frequency enables the multiplexing of information. The figure is based on a model originally described by Akam and Kullmann [23]. Two neuronal populations with differing stimulus sensitivities (“tuning curves”) connect to the same post-synaptic neuronal receiver. The mixed neuronal code is decoded via the receiver’s input gain that matches the target stimulus input train. Red: target. Green: distraction.

**Figure 3 biomedicines-10-02317-f003:**
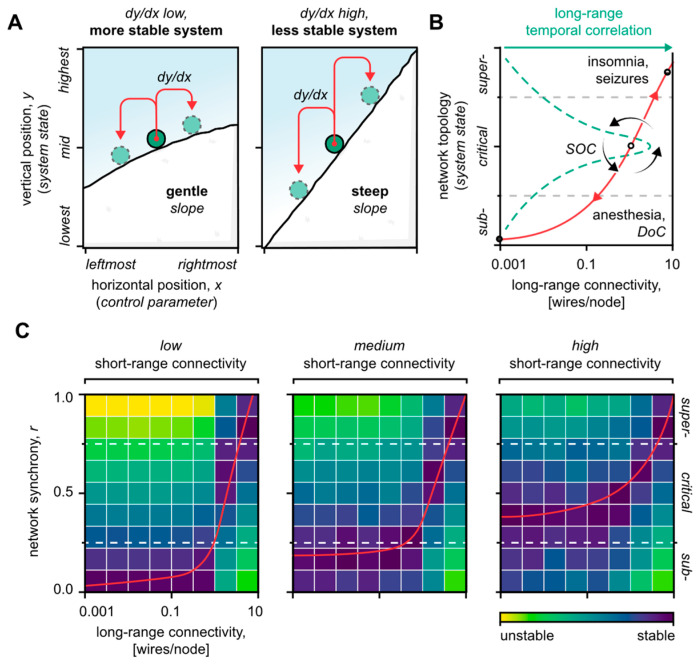
Roles of neuronal connectivity. (**A**) Brain system analogy. The system’s state corresponds to the vertical position of the ball, and the system’s “control parameter” corresponds to the ball’s horizontal position. When the slope steepens, changes to the control parameter cause a greater change in the system’s state (dy/dx increases). The system has destabilized. (**B**) Overview of the topological model. The long-range connectivity is the control parameter that defines the state of the system, represented by a point along the curve. Short-range connections modulate the system dynamic, i.e., the shape of the curve. The curve’s steepness reflects the system’s stability. Sub-critical states are marked by fragmented activity, such as disorders of consciousness (DoC), while super-critical states favor global activity, such as seizures. Near the critical state, the system fluctuates between the local and global modes of operation via self-organized criticality (SOC), counter-balancing the intrinsic topological propensities that pull the system to its extremes (red arrowheads). The brain’s computational power peaks near criticality, mirrored by the peak in the long-range temporal correlation. The long-range temporal correlation curve (green dashes) and disease mappings (black, in-figure text) were adapted with permission from Zimmern (2020) [30]. (**C**) Stability heat-maps based on artificial neural network simulations. Notice the change in the stability map as short-range connectivity decreases. Importantly, as short-range connectivity weakens, the critical regime is destabilized in favor of especially sub-critical system states. Figure adapted from the authors’ previous work [31].

**Figure 4 biomedicines-10-02317-f004:**
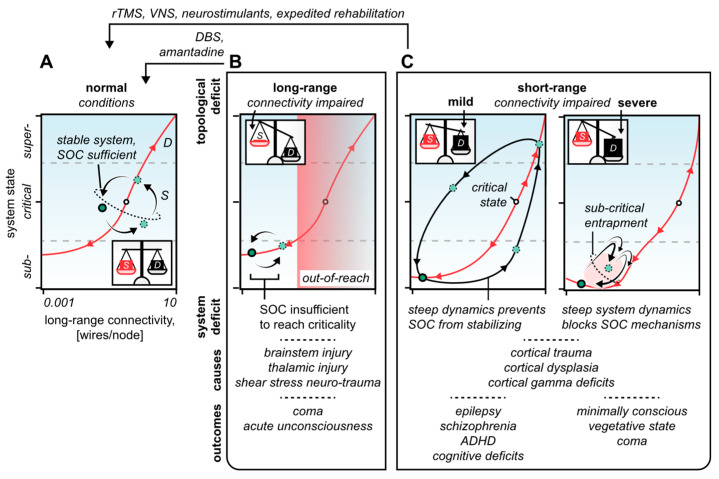
Summary of the hypothesis. (**A**) Under normal conditions, the SOC mechanisms perfectly balance out the system’s topological propensities (see inset). Thus, the system manages to maneuver the critical state to facilitate local (sub-) and global (super-) neurocomputations. (**B**) When long-range connections are impaired, such as by brainstem shear stress, the system’s SOC mechanisms are acutely voided (see inset). This shifts the system toward sub-criticality to an extent that corresponds to the severity of the long-range neuronal injury. Transient loss of consciousness occurs in complete, but reversible, disruption of the long-range neuronal connectivity. In irreversible cases, the extent of long-range injury corresponds to the depth of unconsciousness, ranging from minimally conscious to fully comatose states. Rehabilitation strategies include thalamic neuromodulation, e.g., through DBS or neurostimulant pharmacotherapy, to augment the thalamocortical loop interactions. (**C**) When short-range connections are impaired, such as by a neuro-traumatic cortico-dysfunction, the system’s topological propensities are intensified (see insets). In mild cases, this makes the system fluctuate between extreme dynamical regimes because the SOC mechanisms fail to stabilize near the critical state. In severe cases, the system is effectively trapped in a sub-critical trough in which SOC is completely blocked by the system’s now intense topological propensities. Potential therapeutic strategies include repeated TMS to potentiate cortical short-range connectivity or expedited acute rehabilitation to reduce neuroplastic synaptic depression in neuro-traumatic patients. S, SOC mechanisms; D, the system’s topological propensities; rTMS, repeated transcranial magnetic stimulation; VNS, vagus nerve stimulation; DBS, deep brain stimulation; ADHD, attention deficit hyperactivity disorder. Insets show the balances between the system’s SOC mechanisms and its topological propensities.

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
