# Peer review of "Therapeutic Neuromodulation toward a Critical State May Serve as a General Treatment Strategy"

_biomedicines, 2022, doi:10.3390/biomedicines10092317_

Round 1

Reviewer 1 Report

Interesting paper looking at the role of neural networks in neurologic disease states. The authors go through connected nodes and a general strategy for neural modulation. 

Overall the approach is solid and figures convincing. 

The paper would benefit from expanded discussion regarding the role of neural networks in post op urinary retention PMID: 34507288. 

The authors need to do a better job describing real world implications for neurotrauma and stroke. Nice theoretical concept but fails to show step by step implementation for real world problems.

If the above concept addressed and reference included, paper could be of interest to broader audience.

Author Response

Dear reviewer,

Please find attached our response to your review report.

Best,

Arvin et al.

Reviewer 2 Report

The manuscript by Arvin et al. (Manuscript ID: biomedicines-1849012) on " Therapeutic neuromodulation toward a critical state may serve as a general treatment strategy" aimed to review how a neural network’s functional topology shapes the neural activity that pervades it. The manuscript focuses on the continuous reorganization of neural connectivity around a critical state through brain wave interactions. The authors also examined how network topology and critical dynamics might combine to spur human neuropathology. In my view, the manuscript is written well and updated with all the required information. I have a few comments that may help to improve the quality of the manuscript as follows.

-      The abstract needs to be reorganized and updated summarizing all key aspects of the manuscript.

-       In the introduction section, the authors should explain more about therapeutic neuromodulation with its advantages.

-   Several research studies as well as review papers published on neural networks and neural activity. What are the innovative advancements provided by the authors in this review? The authors should explain this in the introduction section of the manuscript.

-       The manuscript needs to be well polished. The language should be lucid.

-       It will be better if the authors could add at least one figure that describes the overall conclusion of this hypothesis.

Author Response

(The authors gave the same response as above.)

Round 2

Reviewer 1 Report

Accept

Author Response

We thank Reviewer 1 for their acceptance of our manuscript.

Best,

Simon and colleagues.